# Biosynthesis of α-Bisabolol by Farnesyl Diphosphate Synthase and α-Bisabolol Synthase and Their Related Transcription Factors in *Matricaria recutita* L.

**DOI:** 10.3390/ijms24021730

**Published:** 2023-01-15

**Authors:** Yuling Tai, Honggang Wang, Ping Yao, Jiameng Sun, Chunxiao Guo, Yifan Jin, Lu Yang, Youhui Chen, Feng Shi, Luyao Yu, Shuangshuang Li, Yi Yuan

**Affiliations:** School of Life Science, Anhui Agricultural University, Hefei 230036, China

**Keywords:** German chamomile, α-bisabolol, *MrFPS*, *MrBBS*, transcription factor

## Abstract

The essential oil of German chamomile (*Matricaria recutita* L.) is widely used in food, cosmetics, and the pharmaceutical industry. α-Bisabolol is the main active substance in German chamomile. Farnesyl diphosphate synthase (FPS) and α-bisabolol synthase (BBS) are key enzymes related to the α-bisabolol biosynthesis pathway. However, little is known about the α-bisabolol biosynthesis pathway in German chamomile, especially the transcription factors (TFs) related to the regulation of α-bisabolol synthesis. In this study, we identified *MrFPS* and *MrBBS* and investigated their functions by prokaryotic expression and expression in hairy root cells of German chamomile. The results suggest that MrFPS is the key enzyme in the production of sesquiterpenoids, and MrBBS catalyzes the reaction that produces α-bisabolol. Subcellular localization analysis showed that both MrFPS and MrBBS proteins were located in the cytosol. The expression levels of both *MrFPS* and *MrBBS* were highest in the extension period of ray florets. Furthermore, we cloned and analyzed the promoters of *MrFPS* and *MrBBS*. A large number of *cis*-acting elements related to light responsiveness, hormone response elements, and *cis*-regulatory elements that serve as putative binding sites for specific TFs in response to various biotic and abiotic stresses were identified. We identified and studied TFs related to *MrFPS* and *MrBBS*, including *WRKY*, *AP2*, and *MYB*. Our findings reveal the biosynthesis and regulation of α-bisabolol in German chamomile and provide novel insights for the production of α-bisabolol using synthetic biology methods.

## 1. Introduction

German chamomile (*Matricaria recutita* L.; family Asteraceae) is one of the oldest and most widely used medicinal herbs worldwide. Moreover, it is one of the most common herbs native to Europe. Its medicinal value and health effects, such as anti-inflammatory, bacteriostatic, antihypertensive, and antianxiety effects in humans [1,2,3], are due to an abundance of essential oils, especially sesquiterpenoids in flower heads. The key compounds in the essential oil of German chamomile are α-bisabolol, chamazulene, and germacrene D, among others [4].

α-Bisabolol is an unsaturated monocyclic sesquiterpene alcohol; the most important biological and pharmacological activities of α-bisabolol are associated with anti-inflammatory, antibacterial, anti-irritant, and non-allergenic properties. Therefore, α-bisabolol is used in a vast range of products that offer protection against repetitive, environment-induced irritation of the skin, such as hand and body lotions, aftershave creams, lipsticks, sun care and after-sun products, and baby care products [5].

Farnesyl diphosphate synthase (FPS) is a key enzyme in the branch point of the sesquiterpenoids biosynthetic pathway. FPS catalyzes the 1′-4 condensation of dimethylallyl diphosphate (DMAPP) and two molecules of isopentenyl diphosphate (IPP) to form farnesyl pyrophosphate (FPP) [6]. IPP is produced via the methylerythritol phosphate pathway (MEP) and the mevalonate (MVA) pathway [7]. FPP is the precursor of all sesquiterpenoids [8] and is converted into sesquiterpenoids by terpene synthase (TPS). Thus, in this pathway, FPS and TPS are the key enzymes in the production of sesquiterpenoids. FPS genes have been identified in various plant species. Yang et al. [9] cloned and analyzed FPS genes from *Anoectochilus roxburghii* and *Anoectochilus formosanus*. Gaffe et al. [10] found that FPS played an important role in the early development (cell division and enlargement) of plant organs in tomato.

α-Bisabolol synthase (BBS) is a sesquiterpene synthase that catalyzes the last step in the synthesis of α-bisabolol [11]. BBS was first identified in *A. kurramensis* Qazilb. and *A. maritima* L. (*AkBOS* and *AmBOS*, respectively), and has since been identified and characterized in *A. abrotanum* [12]. Recently, a BBS gene was identified in the Brazilian Candeia tree and expressed in *Escherichia coli* [13]. Because BBS is an important sesquiterpene synthase, its biosynthesis mechanism and function have received increasing attention in recent years. However, little is known about the α-bisabolol biosynthesis pathway in German chamomile [14], and few studies have investigated the transcription factors (TFs) associated with the regulation of α-bisabolol synthesis.

Therefore, in the present study, we investigated the genes (*MrFPS* and *MrBBS*) responsible for the biosynthesis of α-bisabolol. We identified the candidate genes related to *MrFPS* and *MrBBS* from the transcriptome data of German chamomile, and then cloned and verified their functions by prokaryotic expression and expression in hairy root cultures of German chamomile. Furthermore, we cloned the promoters of *MrFPS* and *MrBBS* and analyzed the TFs associated with the *MrFPS* and *MrBBS* promoters. Our findings are expected to elucidate the biosynthesis and regulation of α-bisabolol and provide novel insights for production of this compound using synthetic biology methods.

## 2. Results

### 2.1. Cloning and Sequence Analysis of MrFPS and MrBBS from German Chamomile

MrFPS and MrBBS are two important enzymes in the biosynthesis pathway of α-bisabolol. Candidate genes encoding *FPS* and *BBS* were identified and cloned from the German chamomile transcriptome database, and named *MrFPS* and *MrBBS*, respectively. The ORF of *MrFPS* was 1032 bp, which was predicted to encode a 343-amino acid protein with a predicted molecular weight of 37.73 kDa and theoretical isoelectric point (pI) of 5.670. The *MrBBS* gene was predicted to contain an ORF of 1719 bp encoding a 572-amino acid protein with a predicted molecular weight of 62.92 kDa and theoretical pI of 5.42 (Figure 1).

### 2.2. Gene Prokaryotic Expression and Enzyme Activity Detection

The enzymatic activity of MrFPS and MrBBS was estimated by measuring the enzymatic activity of the fluid obtained from *E. coli* BL21 (DE3) pLysS cells transformed with pEASY-Blunt-*MrFPS* and pEASY-Blunt-*MrBBS*. Because the MrFPS reaction product FPP is difficult to detect, we used CIAP to convert FPP into farnesol, and determined this by GC–MS. A peak at 40.76 min was observed in the GC–MS profile of the MrFPS reaction product, and no such peak was observed in controls. MS analysis of the MrFPS reaction product was consistent with farnesol (dephosphorylated from FPP). Moreover, the catalytic MrBBS reaction products were also analyzed by GC–MS, and a peak was detected at 40.1 min in the GC profile. MS analysis of the MrBBS reaction product was consistent with bisabolol, whereas no corresponding peaks were observed in the control. These results demonstrate that MrFPS catalyzed the conversion of the specific substrates (IPP and DMAPP) into FPP, and MrBBS catalyzed the conversion of FPP into α-bisabolol (Figure 1).

### 2.3. Overexpression of MrFPS and MrBBS in Hairy Root Cultures of German Chamomile

The gene expression levels of *MrFPS* and *MrBBS* in hairy root cells transformed with overexpression vectors were higher than those in wild-type root cells and hairy root cells transformed with the empty vector pCAMBIA1302. Three independent lines that expressed *MrFPS* and *MrBBS* were used for quantitative PCR (qPCR) analysis (Figure 2). Moreover, we used GC–MS to determine the content of volatiles in hairy root cell lines that overexpressed *MrFPS* and *MrBBS.* The *MrFPS*-overexpressing hairy root lines showed accumulation of different kinds of sesquiterpenoids, such as α-guaiene, *cis*-α-bisabolene and α-farnesene. *MrBBS*-overexpressing hairy root lines showed accumulation of α-bisabolol. These compounds were 1.2–12-fold higher in overexpressing hairy root lines than in hairy root cells transformed with pCAMBIA1302 alone (Figure 3). The results suggest that MrFPS may be related to the production of sesquiterpenoids, and MrBBS led to the accumulation of α-bisabolol.

### 2.4. Subcellular Localization Analysis of MrFPS and MrBBS

Sesquiterpene synthesis is believed to occur in the cytosol. In our study, we constructed recombinant vectors 35S: *MrFPS*-GFP and 35S: *MrBBS*-GFP and transferred them into tobacco leaves and protoplasts of *Arabidopsis thaliana*. The results showed that MrFPS and MrBBS in the transfected tobacco leaves were present in the cytosol (Figure 4 and Figure 5). Meanwhile, subcellular localization analysis of MrFPS and MrBBS using protoplasts of *A. thaliana* also showed that these two enzymes were located in the cytosol.

### 2.5. Gene Expression Analysis of MrFPS and MrBBS

The expression levels of *MrFPS* and *MrBBS* were analyzed by qPCR in various organs and flowers at different developmental stages (root [R], stem [S], leaf [L], flower bud [FB], extension period of disk and ray florets [F1 and F2, respectively], disk and ray florets in the initial flowering stage differentiation period [3D and 3R, respectively], disk and ray florets in the full-blossom period [4D and 4R, respectively], and disk and ray florets at the end of flowering [5D and 5R, respectively]). The expression level of *MrFPS* was highest in F2 followed by S and L. The expression level of *MrBBS* was highest in F2, followed by F1 and FB (Figure 6). Moreover, the expression levels of both *MrFPS* and *MrBBS* were higher in 4D than in 4R. Phylogenetic analysis showed that MrBBS belongs to the TPS-a subfamily and is most closely related to α-bisabolol synthetase from *Artemisia annua* (Figure 6 and Appendix A).

### 2.6. Promoter Cloning and cis-Acting Element Analysis of MrFPS and MrBBS

The promoters of *MrFPS* and *MrBBS* were cloned, and *cis*-acting elements of the *MrFPS* and *MrBBS* promoters were predicted using the PlantCARE database. Some key elements that form the core promoter, such as TATA and CAAT boxes, were identified. Additionally, a large number of *cis*-acting elements related to light responsiveness, such as the G-box and GT1-motif, were identified. Hormone response elements, such as ABRE (a *cis*-acting element related to abscisic acid reactivity) and the CGTCA-motif (a *cis*-regulatory element involved in methyl jasmonate signaling) were also identified. Moreover, several *cis*-regulatory elements that serve as putative binding sites for specific TFs in response to various biotic and abiotic stresses in plants were identified, for example, ARE (a *cis*-regulatory element necessary for anaerobic induction), LTR (a *cis*-acting element related to low-temperature response), and MBS (a MYB-binding site associated with drought induction (Table 1 and Table 2). Analysis of the putative TF-binding sites showed that there were 115 TFs associated with the APETALA2/ethylene response factor (AP2/ERF) domain, 15 WRKY TFs, and 11 basic helix-loop-helix factors in the *MrFPS* promoter (Appendix A). There were 34 TFs associated with the AP2/ERF domain, 32 WAKY TFs, and 3 Myb/SANT domain factors in the *MrBBS* promoter (Appendix A).

### 2.7. Dual-Luciferase Reporter Analysis of Promoters and TFs

A total of four TFs (1 *MYB*, 2 *AP2*, and 1 *WRKY*) and 14 TFs (5 *MYB*, 6 *AP2*, and 3 *WRKY*) associated with *MrFPS* and *MrBBS* were identified, respectively (Figure 7). We cloned these TFs from German chamomile and the ORFs of these TFs are shown in Appendix A. To further analyze the functions of *MrFPS* and *MrBBS* in German chamomile, the promoter regions of *MrFPS* and *MrBBS* were successfully cloned and sequenced. The ORFs of the transcription factors *MrWRKYs*, *MrMYBs*, and *MrAP2s* were obtained through PCR amplification. To identify whether the *MrFPS* and *MrBBS* promoters can be bound by WRKYs, MYBs, and AP2s, a dual-LUC assay was performed in *Nicotiana benthamiana* leaves. The experimental group (bacterial suspension containing *MrFPS*, MrBBS, and TFs) showed a larger fluorescence area, but the control group (bacterial suspension containing pGreen-0800 and TFs) did not display a fluorescence area. In addition, the value of Luc/Ren was higher in the experimental group than the control group. Combined with the results of the fluorescence area and the value of Luc/Ren, the dual-luciferase reporter assays results showed that three of the four TFs interacted with the *MrFPS* promoter (*MrWRKY1, MrAP21*, and *MrAP22*), and nine of the 14 TFs interacted with the *MrBBS* promoter (*MrMYB3, MrMYB4, MrAP24, MrAP25, MrAP26, MrAP28, MrWRKY2, MrWRKY3*, and *MrWRKY4*; Figure 8, Appendix A).

## 3. Discussion

German chamomile, belonging to the Asteraceae family, is one of the most widely used aromatic and medicinal herbs worldwide. Essential oil from flower heads is the most widely used product due to its beneficial health and medicinal functions. α-Bisabolol is the main sesquiterpenoid in the essential oil of German chamomile. It has a weak, sweet, floral aroma, and is a colorless liquid and a very lipophilic substance. It is soluble in ethanol and almost insoluble in water [15]. The oxidation products are mainly bisabolol-oxide A and bisabolol-oxide B [16]. Previous researches reported that α-bisabolol possess biological and pharmacological activities (antibacterial, antioxidant, anticancer, anti-inflammatory, and others) [17]. Due to the low toxicity of α-bisabolol, the Food and Drug Administration has classified it as “generally regarded as safe” (GRAS); therefore, α-bisabolol is widely used in cosmetic industries such as skin care lotions and creams [18]. Therefore, essential oil from German chamomile may be considered a new source of α-bisabolol.

The biosynthesis pathway of α-bisabolol belongs to the sesquiterpenoids pathway. The MEP and MVA pathways acting upstream provide IPP, and FPS forms the important intermediate product (FPP) by catalyzing the reaction of DMAPP and two molecules of IPP. Lastly, FPP is converted into various sesquiterpenoids by TPS. In plants, FPS is a branch point enzyme in the sesquiterpenoids pathway that performs an important role in sesquiterpenoids biosynthesis [9]. Therefore, cloning and characterization of the MrFPS genes from German chamomile could be helpful for further studies on the biosynthesis and regulation of sesquiterpenoids. The *TPS* gene family is divided into three classes and seven subfamilies: class I contains the TPS-c, TPS-e/f, and TPS-h subfamilies; class II contains the TPS-d subfamily; and class III contains the TPS-a, TPS-b, and TPS-g subfamilies [19]. A previous report showed that most sesquiterpene synthases belong to the TPS-a subfamily [3,20]. Our result showed that *MrBBS* belongs to the TPS-a subfamily and is the final enzyme in the biosynthesis of α-bisabolol [11]. Our results are consistent with previous reports. Therefore, MrBBS is a key enzyme involved in the production of α-bisabolol, and the regulatory mechanism and function of MrBBS have received much attention in recent years (Figure 9).

We identified genes associated with *FPS* and *BBS* in German chamomile and cloned them; the ORFs of *MrFPS* and *MrBBS* were 1032 and 1719 bp, respectively, and encode 343 and 572 amino acid residues, respectively. In our research, we performed the subcellular localization analysis of MrFPS and MrBBS in transfected tobacco leaves and protoplasts of *A. thaliana*, respectively. The results indicated that these two enzymes were both located in the cytosol. FPS has been reported that was located in the cytosol in many plants. Moreover, BBS belongs to sesquiterpene synthases that are localized in the cytosol. Our results are consistent with previous reports, such as *A. thaliana* [21,22,23], periwinkle [24], and Aquilegia species [25]. In vitro enzyme activity analysis suggested that MrFPS and MrBBS act as a functional FPS and α-bisabolol synthase, respectively. For further investigations, we transformed recombinant plasmids containing the *MrFPS* and *MrBBS* genes into *A. rhizogenes*, and then transfected German chamomile and cultivated hairy roots. We observed different types of sesquiterpenoids (α-guaiene, cis-bisabolene, and α-farnesene) in hairy roots of *MrFPS*-overexpressing lines. Meanwhile, we detected α-bisabolol accumulation in hairy roots of *MrBBS*-overexpressing lines. Our results demonstrate that MrFPS is one of the key enzymes in the biosynthesis of sesquiterpenoids, and MrBBS catalyzes the reaction that produces α-bisabolol. However, the regulatory mechanisms of enzymes and genes associated with α-bisabolol production in German chamomile remain unclear.

Recently, a few TFs related to the regulation of terpene synthesis in plants have been characterized, for example, AaWRKY1, AaERF1, AaYABBY5 [26], AaERF2, AabHLH1, and AabZIP1 [27,28,29]. To our knowledge, no study has characterized TFs involved in the regulation of the α-bisabolol biosynthesis pathway in German chamomile. In the present study, we cloned the promoters of *MrFPS* and *MrBBS* from German chamomile, and analyzed the TFs associated with these two genes. A significant number of *cis*-acting elements involved in light responses were found to be associated with both *MrFPS* and *MrBBS*. In addition, hormone response and biotic and abiotic stress response elements such as ABRE, the CGTCA-motif, ARE, LTR, and MYB were identified.

The AP2/ERF is a plant-specific superfamily of TFs with one or two AP2/ERF domains. These TFs are usually closely associated with several physiological processes such as plant growth and development, secondary metabolite biosynthesis, and resistance to biotic and abiotic stresses [30,31]. Recent studies indicate that many plant species, including eggplant [32], rice [33], and tomato [34], have TFs belonging to the AP2/ERF superfamily. In the present study, we screened and identified 115 and 34 putative transcription factor-binding sites related to AP2/ERF domain factors in the *MrFPS* and *MrBBS* promoters, respectively. Furthermore, we cloned MYB, AP2, and WRKY associated with *MrFPS* and *MrBBS*, and verified them by dual-luciferase reporter assay. These results indicate that these TFs from German chamomile are related to stress resistance, in accordance with previous findings [35]. For example, earlier studies reported that German chamomile has considerable adaptability to a wide range of climates [36,37,38]. However, the regulatory mechanism of resistance needs further study. In our study, we identified and cloned TFs (including MYB, AP2, and WRKY) associated with *MrFPS* and *MrBBS*. The regulatory mechanisms of *MrFPS* and *MrBBS*, and their associated TFs, require further investigation.

## 4. Materials and Methods

### 4.1. Plant Materials

German chamomile flower heads (*Matricaria recutita* L.) were gathered during the flowering season from the experimental farm located in Anhui Agricultural University, Hefei, China. All samples, including various organs and flowers at different developmental stages (detailed in Table 3), were collected and immediately frozen in liquid nitrogen and stored at −80 °C until use. Three biological replicates were used per sample. FPP standard (Sigma, Saint Louis, MI, USA) products were purchased from Sigma. Vectors and *Agrobacterium rhizogenes* were stored in the laboratory.

### 4.2. Total RNA and Genomic DNA Extraction and cDNA Synthesis

Total RNA was extracted from German Chamomile samples using RNAiso Plus (Takara Bio Inc, Dalian, China) and treated with an RNAprep pure plant kit (Tiangen, Beijing, China). The CTAB method was used to extract genomic DNA [39], which was stored at −20 °C until use. A NanoDrop 2000 spectrophotometer (NanoDrop Technologies, Wilmington, USA) and agarose gel electrophoresis were used to determine the quality and concentration of the extracted RNA and DNA. A reverse transcription kit (TransGen Biotech, Beijing, China) was used to reverse transcribe the RNA.

### 4.3. Cloning of MrFPS and MrBBS Genes

Genes related to *MrFPS* and *MrBBS* were identified on the basis of transcriptome data from our laboratory (accession number PRJNA382469 in the NCBI SRA database) [35]. These candidate genes were compared using BLAST with the NCBI database to identify the complete coding sequences (CDSs). The CDSs of *MrFPS* and *MrBBS* were obtained by PCR using specific primers designed with Primer 6 (Appendix A). Amplification products were cloned into the Zero Blunt TOPO vector (Yeasen, Shanghai, China) and then sequenced by General Biological System (Chuzhou, China) Co., Ltd.

### 4.4. Prokaryotic Expression and Enzyme Activity Measurement

To obtain the recombinant plasmids pET32a(+)-MrFPS and pCold TF -MrBBS, the CDSs of *MrFPS* and *MrBBS* were cloned into the pET32a(+) vector (Novagen, NJ, USA) and pCold TF (Takara Bio Inc) via restriction enzyme digestion with *Bam*HI and *Sac*I. After transformation of the recombinant plasmids into *E. coli* BL21 (DE3) pLysS (Invitrogen, CA, USA) [40], target gene expression was induced by isopropyl β-D-thiogalactoside [41]. The induced target proteins were detected by sodium dodecyl sulfate–polyacrylamide gel electrophoresis, and soluble proteins were purified by Ni-nitrilotriacetic acid chromatography. Crude MrBBS enzyme (20 µg) was assayed in a reaction mixture containing 100 µM FPP, 50 mM Tris-HCl (pH 7.5), and 10 mM MgCl_2_ in a total volume of 0.5 mL. The enzyme reaction mixtures were then extracted by normal hexane twice. MrFPS enzyme (0.25 µg) was assayed in a total volume of 0.8 mL reaction mixture containing 3.5 µM Tris-HCl (pH 7.6), 0.04 µM dithiothreitol (DTT), 0.005 µM IPP, 0.004 µM MgCl_2_, and 0.005 µM DMAPP, and then 50 µL of 3 mol/L HCl was added to terminate the reaction after incubation at 30 °C for 30 min.

### 4.5. Gas Chromatography-Mass Spectrometry (GC–MS) Analysis of MrFPS and MrBBS Enzyme Reactions

Because FPP is difficult to detect, we firstly used calf intestine alkaline phosphatase (CIAP; Catalog: CP8531-1000U; Coolaber, Beijing, China) to dephosphorylate FPP into farnesol. A 6 µL volume of CIAP Reaction Buffer, 6 µL CIAP, and 48 µL ddH_2_O were mixed to 60 µL [42]. After addition of 20 µL of the above mixture, incubation was performed for 30 min. The experiment was repeated twice. Finally, reaction products were extracted and analyzed by GC–MS using an Agilent 7000B instrument (Agilent, California, USA). The products of MrBBS catalysis were also analyzed by GC–MS. Extracts from *Escherichia coli* BL21 (DE3) pLysS containing empty vector or deactivated enzyme were used as controls. The flow rate of nitrogen was 1.0 mL/min, the injector temperature was 250 °C, and the oven temperature was programmed to increase from 40 °C to 250 °C at 10 °C/min [4]. Volatile components were identified by comparing with spectra in the NIST (National Institute of Standards and Technology) database.

### 4.6. Validation of Transgenic German Chamomile Hairy Roots

To investigate the function of *MrFPS* and *MrBBS*, the recombinant plasmids *pCAMBIA1302-MrFPS* and *pCAMBIA1302-MrBBS* were transformed into *Agrobacterium rhizogenes*, and then transfected into German chamomile for cultivation of hairy roots [39]. One-month-old German chamomile seedlings were excised and wounds were infected with *A. rhizogenes.* The explants were cultivated on MS medium for co-cultivation with bacteria, and then on B5 medium containing cefotaxime. The selection of transformed roots was performed after 6 weeks. *A. rhizogenes* contains the Ri plasmid, and only two open reading frames >300 base pairs (bp) were found in the Ri sequence, namely rolB and rolC genes. This region is a specific gene sequence in the T-DNA region of the Ri plasmid, which has properties of maintaining hairy root morphology and growth. To verify integration of the T-DNA during German chamomile hairy root formation, according to the rolB rolC gene sequence reported previously [39], the presence of *rolB* (770 bp) and *rolC* (540 bp) was detected by PCR, and the gene expression levels of *MrFPS* and *MrBBS* in hairy root cultures were analyzed by qPCR using total RNA extracted from hairy roots. German chamomile hairy root cells transformed with empty vector (pCAMBIA1302) were used as controls, and the 18S rRNA gene was used as the reference gene. Lastly, 0.3 g of transgenic hairy roots and hairy root cells transformed with empty vector was collected and analyzed by SPME-GC–MS.

### 4.7. Subcellular Localization of MrFPS and MrBBS

To study the expression and subcellular localization of MrFPS and MrBBS, we used green fluorescent protein (GFP) labeling technology. Firstly, the CDSs of *MrFPS* and *MrBBS* were cloned by PCR, and the purified PCR products were, respectively, cloned into vector 35S: GFP1300 using the double restriction enzyme digestion method. The resulting recombinant vectors 35S: *MrFPS*-GFP and 35S: *MrBBS*-GFP were transferred into tobacco by agrobacterium injection. The resulting plasmids were transformed into *A. tumefaciens* EHA105, which was cultivated until the OD_600_ reached 0.8, and then the strains were activated by 3 h of dark treatment. Finally, the activated strains were injected into tobacco leaves (1-month-old), and the infected tobaccos were cultured in darkness for 48 h. The protein location was detected using confocal laser scanning microscopy.

### 4.8. Quantitative PCR Analysis and MrBBS Phylogenetic Analysis

We used qPCR to analyze the gene expression levels of *MrFPS* and *MrBBS* in various organs and flowers at different developmental stages of German chamomile. The synthesized cDNA was amplified by qPCR using a TaKaRa SYBR Green qPCR mix and a Bio-Rad CFX 96™ real-time PCR system (Bio-Rad, Hercules, CA, USA). The 18S rRNA gene was chosen as the reference gene. The primers used for qPCR are listed in Appendix A. The 2^ΔCt^ method [43] was used to calculate the relative expression levels of the *MrFPS* and *MrBBS* genes. Three biological and three technical replicates were used for all qPCR analyses. A phylogenetic tree was constructed using the neighbor-joining method in MEGAX [44,45]. The percentage of replicate trees in which the associated taxa clustered together in bootstrap tests (1000 replicates) is shown next to branches. Evolutionary distances were computed using the p-distance method and are presented as the number of amino acid differences per site. This analysis involved 81 amino acid sequences. All ambiguous positions were removed for each sequence pair (pairwise deletion option).

### 4.9. Cis-Regulatory Elements of Promoters

Genomic DNA of German chamomile was used as template, and primers were designed combining both *MrFPS* and *MrBBS* gene cDNA sequences and closely related species as a reference, namely genomic DNA sequences of *Helianthus annuus* (ftp://ftp.ncbi.nlm.nih.gov (accessed on 2 February 2019); [46]) and *Chrysanthemum nankingense* (http://www.amwayabrc.com/ (accessed on 2 February 2019) [47]). The promoters of *MrFPS* and *MrBBS* were subsequently cloned by the PCR method. The *cis*-regulatory elements of *MrFPS* and *MrBBS* were predicted using promoter analysis software PlantCARE (http://bioinformatics.psb.ugent.be/webtools/plantcare/html/ (accessed on 2 March 2019)) and PlantPAN 3.0 (http://plantpan.itps.ncku.edu.tw/promoter.php (accessed on 2 March 2019)).

### 4.10. Identification and Cloning of Transcription Factors Associated with MrFPS and MrBBS

We performed a correlation analysis of the gene expression levels of *MrFPS* and *MrBBS* with the expression patterns of all TFs in the whole-organ transcriptome database of *M. recutita* L. TFs with correlation coefficient ≥0.9 and *p*-value < 0.05 were retained. Candidate TFs were compared with the NCBI database using BLAST to verify the open reading frames (ORFs). Furthermore, the candidate TFs were cloned by PCR using Tks Gflex™ DNA Polymerase (TaKaRa Biomedical Technology Co., Ltd., Beijing, China) (Appendix A).

### 4.11. Dual-Luciferase Reporter Verification

After PCR and purification, the promoter sequences of *MrFPS* and *MrBBS* were cloned into vector pGreen 0800 using a double enzyme digestion method (*Pst*I and *Bam*HI were selected as the restriction enzymes). The recombinant plasmids were named pGreen 0800-*MrFPS* and pGreen 0800-*MrBBS*, respectively. After validation, the plasmids were transformed into *A. tumefaciens* EHA105, which was cultivated in LB medium containing 100 μg/mL K^+^ and 100 μg/mL tetracycline. The strains were cultured on a large scale until the OD_600_ was between 0.8 and 1.0, and then the bacterial solution was centrifuged at 12,000 rpm for 10 min. The cells were resuspended in an equal volume of half-strength Murashige and Skoog (MS) medium. Bacterial suspension containing pGreen-0800 and TFs was used as controls; bacterial suspension containing *MrFPS*, *MrBBS* and TFs was used as the experimental group. The activated strains were injected into tobacco leaves (1-month-old) after dark treatment for 3 h, and then the infected tobaccos were cultured in darkness for 48–72 h. Protein locations were detected using confocal laser scanning microscopy. A Double Luciferase Reporter Gene Test Kit (Yeasen) and a microplate reader were used to determine the Ren/Luc ratio at 560 nm. The dual-luciferase activity of tobacco protein was calculated using the following formula: (experimental group Luc value—background Luc value)/(experimental group Ren value—background Ren value).

### 4.12. Data Analysis

Data in histograms are means ± standard deviations (SD) from *t*-tests, and error bars indicate SD from three biological replicates. Histograms were prepared using Microsoft Excel 2019 software and GraphPad Prism 8.0.2. Three biological replicates were used per sample. Additional data and materials can be made available upon request. The statistical significance of differences between two groups was considered at *p* < 0.05.

## 5. Conclusions

We identified and cloned genes associated with the *MrFPS* and *MrBBS* genes in German chamomile; the ORFs of *MrFPS* and *MrBBS* were 1032 and 1719 bp, respectively, and encoded 343 and 572 amino acid residues, respectively. Moreover, in vitro enzyme activity analysis and overexpression in hairy root cells of German chamomile revealed that MrFPS and MrBBS have farnesyl diphosphate synthase and α-bisabolol synthase activity, respectively. In addition, we identified and cloned TFs associated with the *MrFPS* and *MrBBS* genes and verified them by dual-luciferase reporter assay. Our findings elucidate the biosynthesis and regulatory mechanism of α-bisabolol and provide insights for the synthesis of this compound using synthetic biology methods.

## Figures and Tables

**Figure 1 ijms-24-01730-f001:**
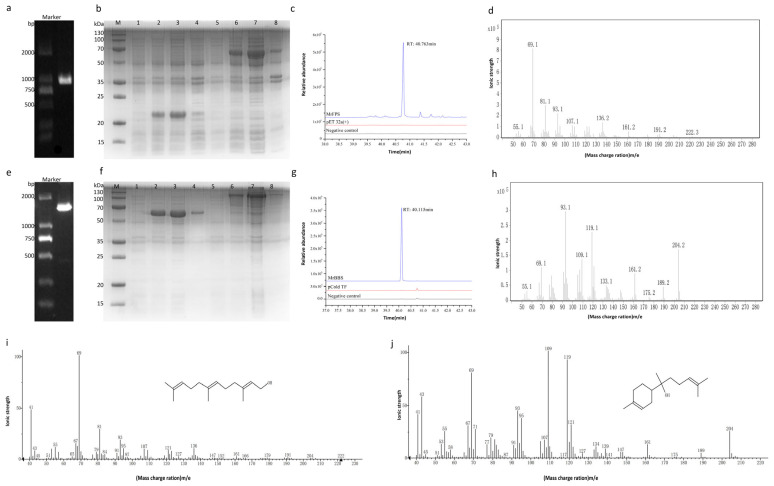
Gene cloning and SDS-PAGE analysis of MrFPS and MrBBS expressed in *E. coli*, and enzyme activity measurement. Note: (**a**): Electrophoreretogram of *MrFPS* amplification; (**b**): SDS-PAGE analysis of MrFPS; (**c**): GC–MS analysis of the enzyme activity of MrFPS; (**d**): Mass spectral analysis of farnesol; (**e**): Electrophoreretogram of *MrBBS* amplification; (**f**): SDS-PAGE analysis of MrBBS; (**g**): GC–MS analysis of the enzyme activity of MrBBS; (**h**): Mass spectral analysis of α-bisabolol; (**i**): Standard mass spectra of farnesol; (**j**): Standard mass spectra of α-bisabolol; b: Lane 1, before IPTG induction of empty vector pet32; lane 2, after IPTG induction of empty vector pet32; lane 3, soluble pet32 protein, lane 4, insoluble pet32 protein, lane 5, before IPTG induction of recombinant protein MrFPS, lane 6, after IPTG induction of recombinant protein MrFPS, lane 7, soluble MrFPS protein, lane 8, insoluble MrFPS protein; f:Lane 1, before IPTG induction of empty vector pet32; lane 2, after IPTG induction of empty vector pCold TF;; lane 3, soluble pCold TF; protein, lane 4, insoluble pCold TF; protein, lane 5, before IPTG induction of recombinant protein MrBBS, lane 6, after IPTG induction of recombinant protein MrBBS, lane 7, soluble MrBBS protein, lane 8, insoluble MrBBS protein.

**Figure 2 ijms-24-01730-f002:**
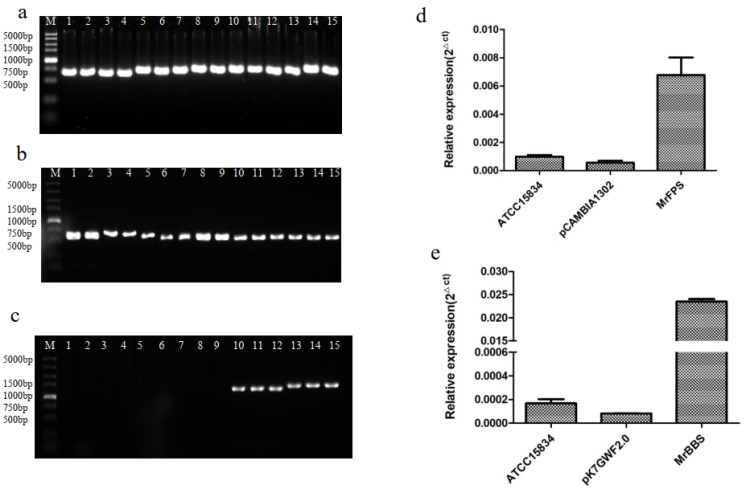
PCR analysis of DNA from *MrFPS* (**a**) and *MrBBS* (**b**) transgenic hairy roots using primers specific to the *rol B* and *rolC* genes. (**c**) PCR analysis of DNA from transgenic hairy roots using primers specific to *MrFPS* and *MrBBS.* Note: Lane M, markers; c: lanes 1–3, ATCC15834 DNA; lanes 4–6, pK7GWF2.0 DNA; lanes 7–9, Pcambia1302; lanes 10–12, PCR products of *MrBBS* from putative transformant; lanes 11–15, PCR products of *MrFPS* from putative transformants; qPCR analysis on DNA from wild-type roots and overexpression of *MrFPS* in hairy roots (**d**); qPCR analysis on DNA from wild-type roots and overexpression of *MrBBS* in hairy roots (**e**).

**Figure 3 ijms-24-01730-f003:**
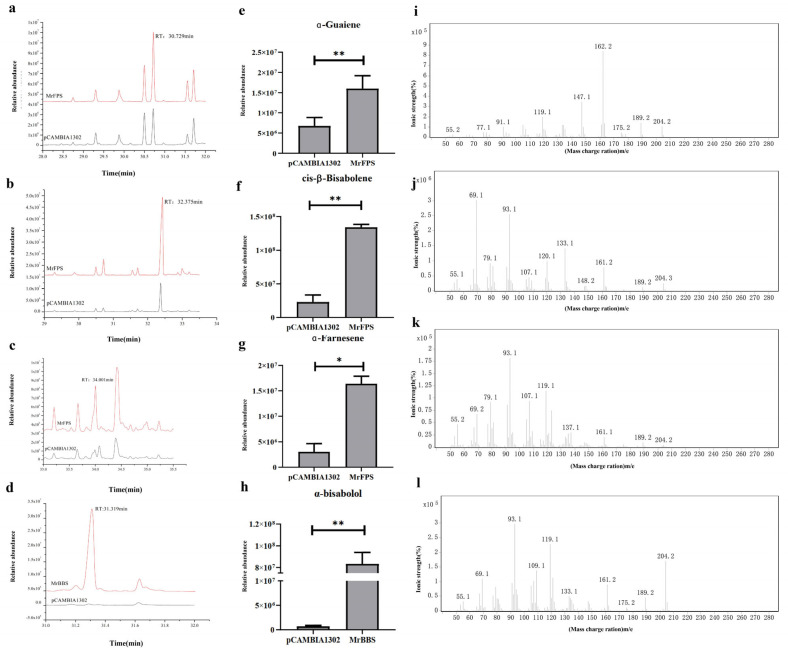
GC–MS analysis of overexpression of *MrFPS* (**a**–**c**) and *MrBBS* (**d**) in German chamomile hairy roots. Relative content analysis of α-guaiene, *cis*-α-bisabolene, α-farnesene, and α-bisabolol in overexpressing German chamomile hairy roots (**e**–**h**). (**i**) Mass spectrometry analysis of α-guaiene. (**j**) Mass spectrometry analysis of *cis*-α-bisabolene. (**k**) Mass spectrometry analysis of α-farnesene f. (**l**) Mass spectrometry analysis of α-bisabolol. Note: pCAMBIA1302 indicates hairy roots transformed with empty vector. Error bars are shown with three biological replicates (*t*-test). One asterisk (*) indicates a significant difference (0.01 < *p* < 0.05) and ** indicate a very significant difference (*p* < 0.01).

**Figure 4 ijms-24-01730-f004:**
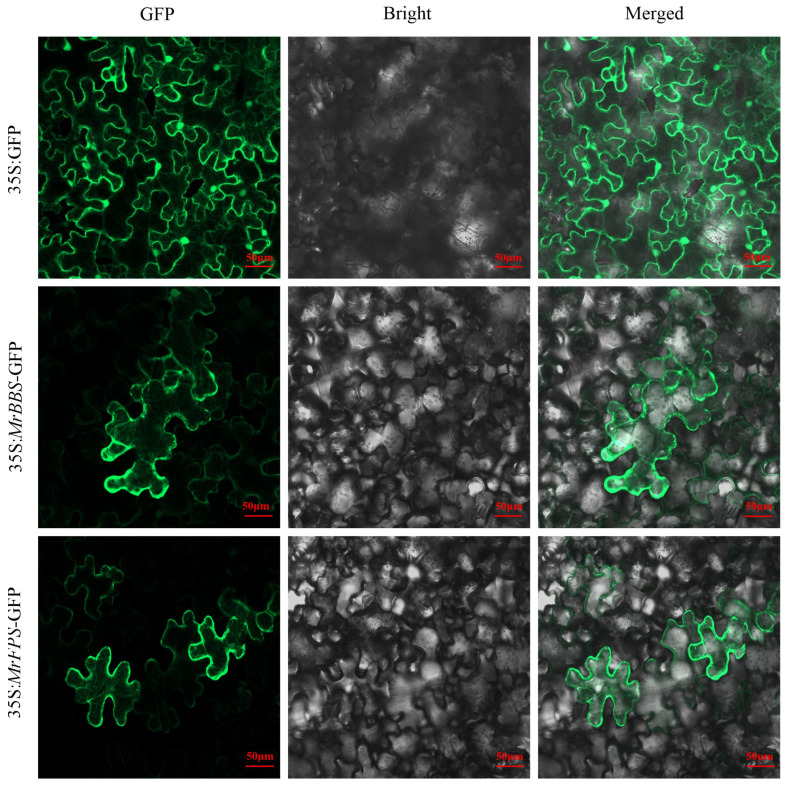
Subcellular localization of MrFPS and MrBBS in tobacco. All samples were visualized using confocal microscopy. GFP means GFP fluorescence; Bright means Light field; Merged means superposition of fluorescence. Control vector (35S: GFP) and recombinant vectors (35S: *MrFPS*-GFP and 35S: *MrBBS*-GFP) were expressed in protoplasts of tobacco.

**Figure 5 ijms-24-01730-f005:**
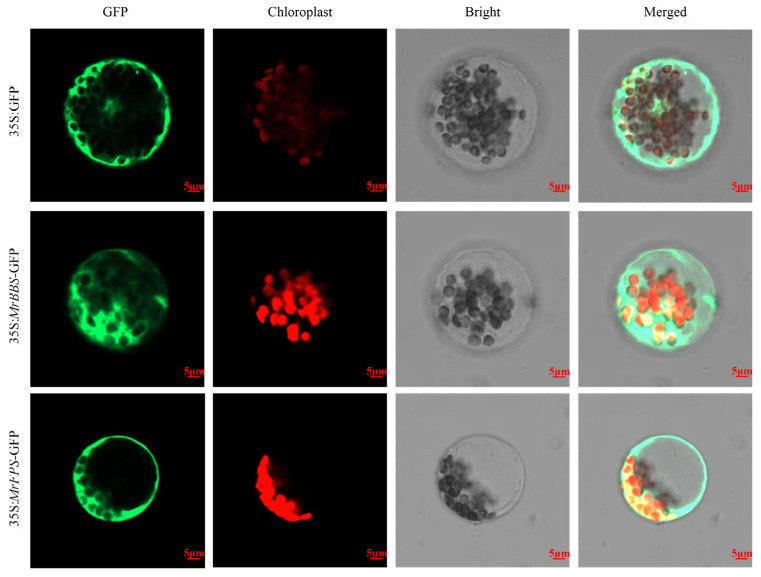
Subcellular localization of MrFPS and MrBBS in protoplasts of *Arabidopsis thaliana.* All samples were visualized using confocal microscopy. GFP means GFP fluorescence; Chlorophyll means Chlorophyll fluorescence; Bright means Light field; Merged means superposition of fluorescence. Control vector (35S: GFP) and recombinant vectors (35S: *MrFPS*-GFP and 35S: *MrBBS*-GFP) were expressed in protoplasts of *Arabidopsis thaliana*.

**Figure 6 ijms-24-01730-f006:**
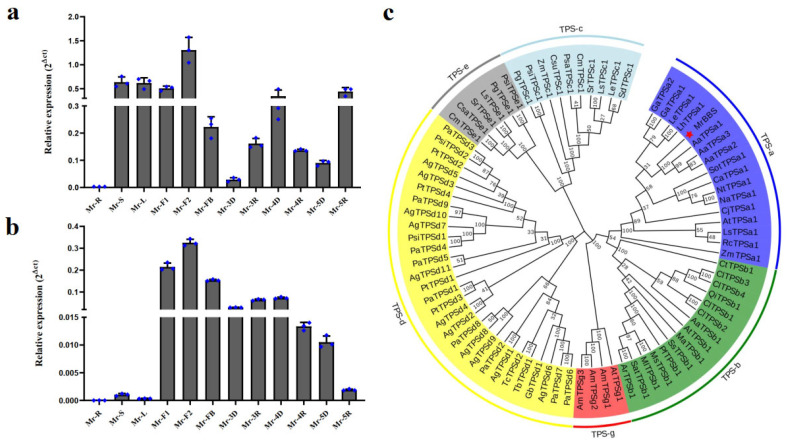
Expression patterns of *MrFPS* (**a**) and *MrBBS* (**b**) in German chamomile, and phylogenetic tree analysis of *MrBBS* in German chamomile (**c**). Each point represents one independent measurement.

**Figure 7 ijms-24-01730-f007:**
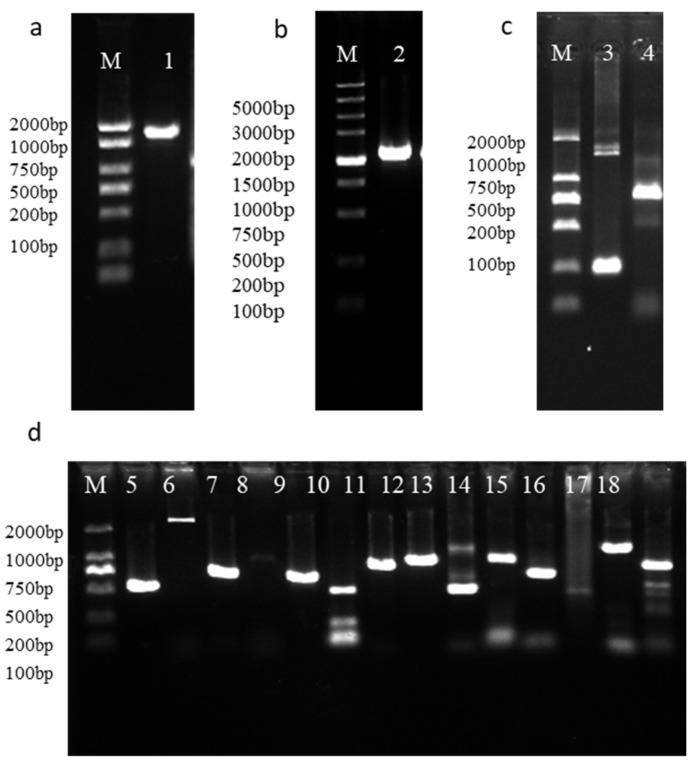
PCR amplification of TFs related to *MrFPS* (**a**–**c**) and *MrBBS* (**d**). Lane M, markers; lane 1, *MrMYB1*; lane 2, *MrAP22*; lane 3, *MrWRKY1*; lane 4, *MrAP21*; lane 5, *MrMYB2*; lane 6, *MrMYB3*; lane 7, *MrMYB4*; lane 8, *MrMYB5*; lane 9, *MrMYB6*; lane 10, *MrAP23*; lane 11, *MrAP24*; lane 12, *MrAP25*; lane 13, *MrAP26*; lane 14, *MrAP27*; lane 15, *MrAP28*; lane 16, *MrWRKY2*; lane 17, *MrWRKY3*; lane 18, *MrWRKY4*.

**Figure 8 ijms-24-01730-f008:**
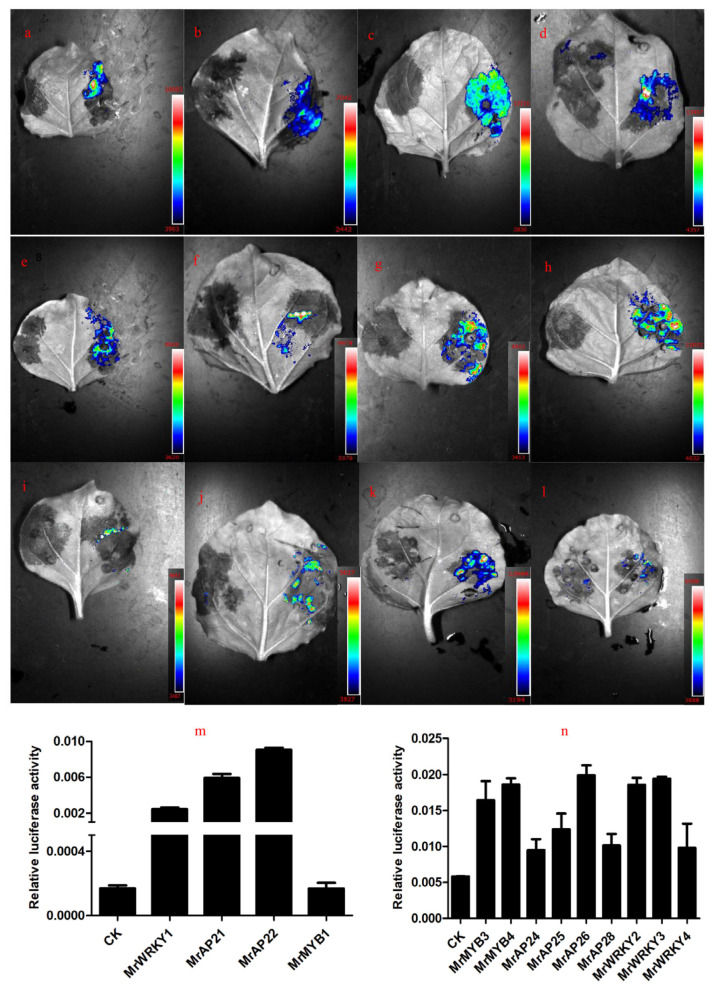
Dual-luciferase reporter analysis of *MrFPS* and *MrBBS* promoters and transcription factors. (**a**–**c**) Double luciferase activity of the *MrFPS* promoter with *MrWRKY1*, *MrAP21*, and *MrAP22*. (**d**–**l**) Double luciferase activity of the *MrBBS* promoter with *MrMYB3*, *MrMYB4*, *MrAP24*, *MrAP25*, *MrAP26*, *MrAP28*, *MrWRKY2*, *MrWRKY3*, and *MrWRKY4*. (**m**) Analysis of the double luciferase activity of the *MrFPS* promoter with *MrWRKY1*, *MrAP21* and *MrMYB1*, and *MrAP22*. (**n**) Analysis of the double luciferase activity of the *MrBBS* promoter with *MrMYB3*, *MrMYB4*, *MrAP24*, *MrAP25*, *MrAP26*, *MrAP28*, *MrWRKY2*, *MrWRKY3*, and *MrWRKY4*. Note: Bacterial suspension containing pGreen-0800 and transcription factors was mixed for controls. Error bars reflect three biological replicates. The color difference represents the intensity of the fluorescence value.

**Figure 9 ijms-24-01730-f009:**
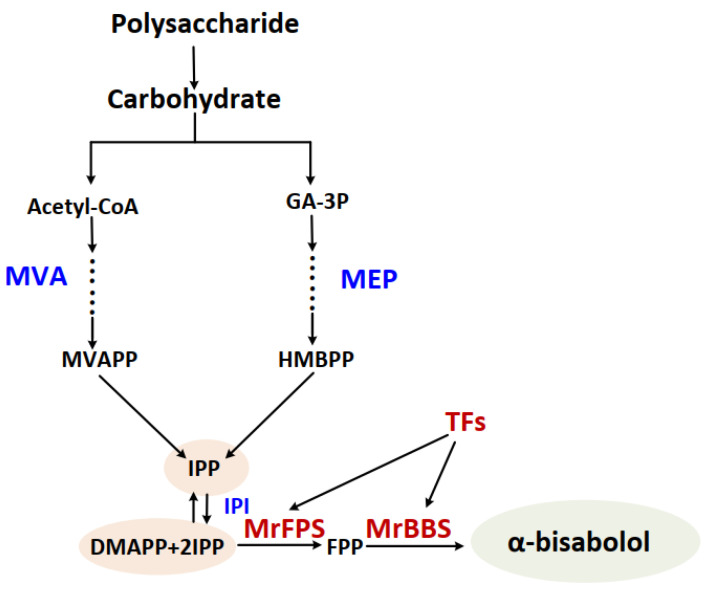
The biosynthesis pathway of α-bisabolol in German chamomile.

**Table 1 ijms-24-01730-t001:** Prediction of the cis-acting elements of the promoter region of the *MrFPS* gene in German chamomile.

Site Name	Position Strand	Sequence	Function
A-box	−1706	CCGTC	Cis-acting regulatory element
ABRE	−587/−1185 −1393/−1690	ACGTG CACGTG/ CGCACGTGTC	Cis-acting element involved in abscisic acid responsiveness
ARE	−916	AAACCA	Cis-acting regulatory element essential for the anaerobic induction
CAAT-box	−769/−1060/−1927 −864/−1417	CCAAT CAAT/ CAAAT	Common cis-acting element in promoter and enhancer regions
CAT-box	−595	GCCACT	Cis-acting regulatory element related to meristem expression
CCGTCC-motif	−1706	CCGTCC	
CGTCA-motif	−1824	CGTC	Cis-acting regulatory element involved in the MeJA responsiveness
G-Box	−1392/−1689/−1392 −1073/−1779	CACGTG	Cis-acting regulatory element involved in light responsiveness
GC-motif	−85	CCCCCG	Enhancer-like element involved in anoxic-specific inducibility
I-box	−1842	CGATAAGGCG	Part of a light-responsive element
LTR	−888	CCGAAA	Cis-acting element involved in low-temperature responsiveness
P-box	−380/−555	CCTTTTG	Gibberellin-responsive element
Pc-CMA2c	−111	GCCCACGCA	Part of a light-responsive element
STRE	−25/−253	AGGGG	
Sp1	−249/−1592 −1011/−1496	GGGCGG	Light-responsive element
TATA-box	−909	TATA	Core promoter element around -30 of transcription start
TGACG-motif	−1699	TGACG	Cis-acting regulatory element involved in the MeJA responsiveness
MYB	−1308	CAACCA	
MYB recognition site	−609	CCGTTG	
MYC	−402/−1835/−1892	CATGTG	
Myb-binding site	−1308	CAACAG	
ABRE3a	−586/−1184	TACGTG	
Unnamed__1	−932/−150/−1113	CGTGG	
Unnamed__2	−803/−1456 −1356/−1499	CCCCGG	
Unnamed__4	−267/−278/−893 −978/−1813/−1905	CTCC	
W box	−694	TTGACC	
as-1	1699	TGACG	
dOCT	−699	CTCGGATC	
re2f-1	−95	GCGGGAAA	

**Table 2 ijms-24-01730-t002:** Prediction of the cis-acting elements of the promoter region of the *MrBBS* gene in German chamomile.

Site Name	Position Strand	Sequence	Function
A-box	−661	CCGTCC	Cis-acting regulatory element
ABRE	−5	ACGTG	Cis-acting element involved in abscisic acid responsiveness
ARE	−296	AAACCA	Cis-acting regulatory element essential for anaerobic induction
AuxRR-core	−246	GGTCCAT	Cis-acting regulatory element involved in auxin responsiveness
CAAT-box	−281/−866 −294/−411	CCAAT/CAAT	Common cis-acting element in promoter and enhancer regions
CGTCA-motif	−331	CGTCA	Cis-acting regulatory element involved in the MeJA responsiveness
G-Box	−499	CACGTT	Cis-acting regulatory element involved in light responsiveness
I-box	−377	GGATAAGGTG	Part of a light-responsive element
LTR	−147	CCGAAA	Cis-acting element involved in low-temperature responsiveness
MBS	−594	CAACTG	MYB-binding site involved in drought-inducibility
O_2_-site	−94	GATGACATGG	Cis-acting regulatory element involved in zein metabolism regulation
P-box	−326	CCTTTTG	Gibberellin-responsive element
TATA-box	−544	TATA	Core promoter element around -30 of transcription start
TCT-motif	−257	TCTTAC	Part of a light-responsive element
Unnamed__1	−40	CGTGG	
Unnamed__4	−602/−134	CTCC	
W box	−320/−344	TTGACC	
as-1	−782/−513	TGACG	
STRE	−82	AGGGG	
CCGTCC-motif/ CCGTCC-box	−661	CCGTCC	
Myb	−594	CAACTG	
TCA	−858	TCATCTTCAT	
MYC	−480	CATTTG	

**Table 3 ijms-24-01730-t003:** Tissues and different flower developmental stages of German chamomile used in the present study.

Name	Abbreviation
Root	Mr-R
Stem	Mr-S
Leaf	Mr-L
Flower bud differentiation period	Mr-FB
Extension period of disk florets	Mr-F1
Extension period of ray florets	Mr-F2
Disk florets in the initial flowering stage differentiation period	Mr-3D
Ray florets in the initial flowering stage differentiation period	Mr-3R
Disk florets in the full-blossom period	Mr-4D
Ray florets in the full-blossom period	Mr-4R
Disk florets at the end of flowering	Mr-5D
Ray florets at the end of flowering	Mr-5R

## Data Availability

Not applicable.

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
