# Peer review of "Biosynthesis of α-Bisabolol by Farnesyl Diphosphate Synthase and α-Bisabolol Synthase and Their Related Transcription Factors in Matricaria recutita L."

_ijms, 2023, doi:10.3390/ijms24021730_

Round 1

Reviewer 1 Report

Assay of enzymatic activity in vitro is the key to elucidate the functions of MrFPS and MrBBS. In this paper, the products of the enzymatic reaction were analyzed by the obtained mass spectrum comparing with database. This is controversial. The standard chemicals of farnesol and bisabolol should be used as positive controls. Moreover, the discription of the results of dual-luciferase reporter analysis of promoters and TFs is insufficient, more detailed results should be presented.

Author Response

To reviewer 1

Comments and Suggestions for Authors

Assay of enzymatic activity in vitro is the key to elucidate the functions of MrFPS and MrBBS. In this paper, the products of the enzymatic reaction were analyzed by the obtained mass spectrum comparing with database. This is controversial. The standard chemicals of farnesol and bisabolol should be used as positive controls. Moreover, the discription of the results of dual-luciferase reporter analysis of promoters and TFs is insufficient, more detailed results should be presented.

REPLY: Thank you very much for the suggestions. Firstly, we analyzed the standard mass spectra of farnesol and α-bisabolol in the NIST database, and compared the mass spectra of enzyme products to the standard mass spectra of farnesol and α-bisabolol. The results showed that enzyme products of MrFPS and MrBBS were farnesol and α-bisabolol, respectively. We have added and revised this result in Figure 1. Secondly, we overexpressed MrFPS and MrBBS in hairy root cells of German chamomile, and revealed that MrFPS and MrBBS possess farnesyl diphosphate synthase and α-bisabolol synthase activity, respectively. These results are consistent with the in vitro enzyme activity analysis. In addition, we have added a description of the results of dual-luciferase reporter analysis of promoters and TFs in the revised manuscript.

Reviewer 2 Report

In this study, authors characterized functions of two genes encoding FPS and BBS from German chamomile. Authors conducted several biochemical and molecular works. Experiments were nicely done with clear results. I am quite impressed with the results from this manuscript. Excellent. I have some minor comments as follows.

Please describe about chemicals or kits in detail. For example, information for city and country is missing. (Sigma, City, Country) L95

Again (TaKaRa Bio Inc., City, Country) Check them thoroughly in the manuscript.

L135-L140 Check the font size.

L144-145 In this study, authors generated transgenic German chamomile plants by Agrobacterium rhizogenes. In fact, it is not easy to get transgenic plants from the non-model plants. Could you describe about transformation procedures for German chamomile in detail?

In addition, Why did you use Agrobacterium rhizogenes?

How did you obtain Agrobacterium rhizogenes?

For T-DNA integration, authors checked the presence of two genes for rolB and rolC. Could you write about that in detail?

L181-182, In the manuscript, authors mentioned that they used the transcriptome of German chamomile from the previous study. However, why did authors design primers based on the closely related species?

L227 Supplementary Figure 1 should be Figure 1. The quality of image should be improved.

L243 This should be Figure 2. The image sizes of Figure 2 should be magnified.

L266 Supplementary Figure 2 should be Figure 3.

L271 Supplementary Figure 3 should be Figure 4.

L274 Figure 2 should be Figure 5. The image sizes should be magnified.

L288 Figure 3 should be Figure 6.

L293 Figure 4 should be Figure 7.

L310 Figure 5 should be Figure 8.

L344 Supplemental Figure 4. Should be Figure 9.

L351 Figure 6 should be Figure 10.

L378 Figure 7 should be Figure 11.

I think it is desirable to show all figures including supplementary figures in the manuscript.

The manuscript was not formatted to IJMS. Please reformat the manuscript.

Discussion was relatively short. I would like to ask authors to discuss more about their results with previous studies. It is also desirable to describe their results associated with possible applications.

Overall, I feel that the purpose and results of this study are excellent for publication. However, the manuscript was prepared for short time. Please read and revise their manuscript carefully. English editing might be helpful.

Author Response

To reviewer 2

Comments and Suggestions for Authors

In this study, authors characterized functions of two genes encoding FPS and BBS from German chamomile. Authors conducted several biochemical and molecular works. Experiments were nicely done with clear results. I am quite impressed with the results from this manuscript. Excellent. I have some minor comments as follows.

Please describe about chemicals or kits in detail. For example, information for city and country is missing. (Sigma, City, Country) L95

Again (TaKaRa Bio Inc., City, Country) Check them thoroughly in the manuscript.

REPLY: Thank you very much for the suggestions. All have been checked and corrected in the revised manuscript. 

L135-L140 Check the font size.

REPLY: Thank you very much for suggestion. We have revised the manuscript accordingly.

L144-145 In this study, authors generated transgenic German chamomile plants by Agrobacterium rhizogenes. In fact, it is not easy to get transgenic plants from the non-model plants. Could you describe about transformation procedures for German chamomile in detail?

REPLY: We established a transformation system for German chamomile hairy roots using Agrobacterium rhizogenes, and published this work in 2020 (Chengcheng Ling et al., Plant Science, 2020). We have cited this article and added the method for generation of German chamomile hairy roots in the revised manuscript.

In addition, Why did you use Agrobacterium rhizogenes?

REPLY: Agrobacterium rhizogenes is the causative agent of hairy root disease in plants. It has been demonstrated that the Ri plasmid present in A. rhizogenes transforms plant cells by introducing its T-DNA into the genome of plant cells. The transformed plant cells then grow to form hairy roots. In addition, an important feature of A. rhizogenes-induced roots is their unique ability to grow in vitro in the absence of exogenous plant growth regulators. The Agrobacterium-mediated method can further accelerate the genetic transformation of hairy roots.

How did you obtain Agrobacterium rhizogenes?

REPLY:  Agrobacterium rhizogenes was kept in the laboratory of our research group.

For T-DNA integration, authors checked the presence of two genes for rolB and rolC. Could you write about that in detail?

REPLY: Thank you very much for suggestion. We revised in the manuscript accordingly.

L181-182, In the manuscript, authors mentioned that they used the transcriptome of German chamomile from the previous study. However, why did authors design primers based on the closely related species?

REPLY: Thank you. Our research group sequenced the transcriptome of German chamomile, hence we cloned the cDNA sequences of MrFPS and MrBBS using the transcriptome of German chamomile. To date, there are no published genome sequences for German chamomile. Thus, we cloned the promoters of MrFPS and MrBBS based on homologs in closely related species. In order to describe this more clearly, we have changed and this section in the revised manuscript.

L227 Supplementary Figure 1 should be Figure 1. The quality of image should be improved.

REPLY: Thank you very much for suggestion. We revised in the manuscript.

L243 This should be Figure 2. The image sizes of Figure 2 should be magnified.

REPLY: Thank you very much for suggestion. We revised in the manuscript.

L266 Supplementary Figure 2 should be Figure 3.

REPLY: Thank you very much for suggestion. We revised in the manuscript.

L271 Supplementary Figure 3 should be Figure 4.

REPLY: Thank you very much for suggestion. We revised in the manuscript.

L274 Figure 2 should be Figure 5. The image sizes should be magnified.

REPLY: Thank you very much for suggestion. We revised in the manuscript.

L288 Figure 3 should be Figure 6.

REPLY: Thank you very much for suggestion. We revised in the manuscript.

L293 Figure 4 should be Figure 7.

REPLY: Thank you very much for suggestion. We revised in the manuscript.

L310 Figure 5 should be Figure 8.

REPLY: Thank you very much for suggestion. We revised in the manuscript.

L344 Supplemental Figure 4. Should be Figure 9.

REPLY: Thank you very much for suggestion. We revised in the manuscript.

L351 Figure 6 should be Figure 10.

REPLY: Thank you very much for suggestion. We revised in the manuscript.

L378 Figure 7 should be Figure 11.

REPLY: Thank you very much for suggestion. We revised in the manuscript.

I think it is desirable to show all figures including supplementary figures in the manuscript.

REPLY: Thank you very much for suggestion. We revised and merged the supplementary figures in the revision.

The manuscript was not formatted to IJMS. Please reformat the manuscript.

REPLY: Thank you very much for suggestion. We revised the full text to improve the quality of this revision.

Discussion was relatively short. I would like to ask authors to discuss more about their results with previous studies. It is also desirable to describe their results associated with possible applications.

REPLY: Thank you very much for suggestion. We revised in the manuscript.

Overall, I feel that the purpose and results of this study are excellent for publication. However, the manuscript was prepared for short time. Please read and revise their manuscript carefully. English editing might be helpful.

REPLY: Thank you very much for suggestion. We asked the experts to edit English writing of the manuscripts and revised in the manuscript.
